# Cryptococcal Meningitis in HIV-Negative Patients: A 12-Year Single-Center Experience in China

**DOI:** 10.3390/jcm12020515

**Published:** 2023-01-08

**Authors:** Yu Huang, Xiaozhi Jin, Faling Wu, Tongtong Pan, Xiaodong Wang, Dazhi Chen, Yongping Chen

**Affiliations:** 1Zhejiang Provincial Key Laboratory for Accurate Diagnosis and Treatment of Chronic Liver Diseases, The First Affiliated Hospital of Wenzhou Medical University, Wenzhou 325035, China; wshuangyu@126.com (Y.H.); jinxiaozhi1984@live.cn (X.J.); wufaling@wmu.edu.cn (F.W.); ptt_wmu@163.com (T.P.); xdwang0001@163.com (X.W.); 2Department of Infectious Diseases, The First Affiliated Hospital of Wenzhou Medical University, Wenzhou 325035, China; 3Department of Clinical Medicine, Hangzhou Medical College, Hangzhou 310053, China

**Keywords:** cryptococcal meningitis, HIV-negative, risk factors, alcohol abuse, kidney transplant

## Abstract

Objective: Cryptococcal meningitis (CM) is a not rare condition in HIV-negative patients. Here, we describe the clinical characteristics, possible risk factors, and outcomes of HIV-negative patients with CM. Methods: Medical records from 99 HIV-negative patients with CM admitted to our hospital from 2010 to 2021 were reviewed systematically. We compared the clinical features and outcomes between patients with underlying diseases and otherwise healthy hosts. Results: The 99 HIV-negative CM patients had a mean age at presentation of 56.2 ± 16.2 years, and the female-to-male ratio was 77:22. A total of 52 (52.5%) CM patients had underlying conditions, and 47 patients (47.5%) had no underlying conditions. Kidney transplant represented the most frequent underlying condition (11.1%), followed by rheumatic disease (10.1%) and hematological diseases (9.1%). Compared to patients without underlying conditions, those with underlying conditions had significantly more fever, more steroid therapy, higher serum creatinine, and lower albumin, IgG, hemoglobin, and platelets (*p* < 0.05 for each). CM patients without underlying conditions had significantly more alcohol abuse than those with underlying conditions (31.9% vs. 9.6%, *p* = 0.011). By logistic regression analysis, male gender (OR = 3.16, *p* = 0.001), higher CSF WBC (OR = 2.88, *p* = 0.005), and protein (OR = 2.82, *p* = 0.002) were significantly associated with mortality. Conclusion: Patients with underlying conditions had a similar mortality to patients without underlying conditions. Alcohol abuse was a probable risk factor for CM for previously healthy patients. Male gender, higher CSF WBC, and protein were significantly associated with mortality.

## 1. Introduction

Cryptococcal meningitis (CM) is the leading cause of fungal meningitis in humans worldwide, especially in areas with a high prevalence of human immunodeficiency virus (HIV) [1,2]. CM is the second leading cause of HIV-related mortality, with the majority of deaths occurring in sub-Saharan Africa, where up to 60% of people with HIV reside [3]. In non-HIV endemic areas, CM is also present in a growing number of patients with natural or iatrogenic immunosuppression, with high rates of death despite therapy [4]. In the United States, deaths from non-HIV-related CM now account for approximately one-quarter of CM-related hospitalizations and one-third of CM-related deaths [5].

Nationally representative data show a decline in cryptococcal infections in developed nations with the advent of highly active antiretroviral therapy [5,6]. However, HIV-negative patients with cryptococcosis experience delayed diagnosis and have higher mortality rates than HIV-infected patients, affecting nearly 50% of all patients [7]. A US study reported the 90-day mortality rate of HIV-negative patients with CM as 27%, which is higher than that of HIV-positive patients, and a significant proportion of patients have no underlying conditions [8]. Despite some studies of CM in HIV-negative patients, the data from Asia are limited, especially in the past 10 years. In addition, the immunocompetent population accounts for an increasing proportion of morbidity, and much less has been written on the effects of lifestyle on the risk of CM in these patients [9,10].

We performed a retrospective observational study to describe the incidence, clinical features, prognostic and risk factors, and lifestyle of HIV-negative patients diagnosed with CM at a single center in China.

## 2. Patients and Methods

### 2.1. Patients

Patients hospitalized in the First Affiliated Hospital of Wenzhou University from January 2010 to December 2021 were selected for this retrospective study. Records of all of the patients with CM during a 12-year period were obtained.

This study was approved by the Ethics Committee of the First Affiliated Hospital of Wenzhou Medical University (KY2022-R030). All of the enrolled patients signed informed consent forms. All methods were carried out in accordance with the relevant guidelines and regulations.

### 2.2. Diagnosis of CM

Cerebrospinal fluid (CSF) was routinely sent for complete white blood cell counts and differential counts, glucose, protein, and India ink stain and cultures. A definite diagnosis of CM was determined if the patient met at least one of the following criteria: (1) positive culture of Cryptococcus from CSF, (2) positive India ink smear of CSF centrifuged sediment for Cryptococcus. Immy Latex-Crypto Antigens (Immuno-Mycologics, Inc., Wenzhou, China) were used to perform the antigen assay in the blood or CSF, but not as a diagnostic criterion.

### 2.3. Clinical Profiles

Demographic characteristics of the study population were determined during the admission for incident CM. For each patient, the following data were obtained: demographic information, manifestations, underlying diseases (organ transplantation, autoimmune diseases, hematologic malignancies, primary nephrotic syndrome, cirrhosis, chronic renal failure, and solid malignancies), glucocorticoid therapy (defined as oral prednisone for at least three months in the past year), diabetes, laboratory data, including serum albumin, immunoglobulin G (IgG), serum creatinine, serum hemoglobin (Hb), white blood cells (WBC), and platelets, results of CSF analyses, magnetic resonance imaging (MRI), antifungal treatments, and patient outcomes. Chronic renal failure was defined as eGFR < 60 mL/min/1.73 m^2^. Patients underwent a lumbar puncture for CSF analysis and MRI at the time of diagnosis, and a repeat lumbar tap was performed if needed. Patient outcomes were short-term mortality, defined as death within 30 days of admission, and long-term mortality, defined as death before December 2021.

In addition, environmental exposures and lifestyle of CM patients were also recorded, including occupation, pet ownership, smoking, alcohol abuse, marijuana use, sleep late, sleep insufficiency, and living in a wet environment. We defined sleep late as going to sleep after 0 a.m., sleep insufficiency as sleeping less than 6 h per day, living in a wet environment as living on the first floor or basement, and alcohol abuse as drinking daily.

### 2.4. Statistical Analysis

Variables are described as the mean plus the standard deviation or proportion. Differences between patients in different groups were compared by the independent-sample *t*-test or chi-square test. Multivariate logistic regression was used to determine the independent clinical variables associated with CM patients who died. A *p*-value of <0.05 was considered statistically significant. All statistical analyses were performed using SPSS 22 (IBM, Cary, NC, USA).

## 3. Results

From January 2010 to December 2021, 106 cases of CM were documented. Among them, there were significantly more cases in non-HIV-infected patients (*n* = 99, 93.4%) than in HIV-infected patients (*n* = 7, 6.6%). There was a significantly increased trend in the incidence of non-HIV-infected patients but no significant trend in HIV-infected cases.

The 99 HIV-negative CM patients had a mean age at presentation of 56.2 ± 16.2 years, and the female-to-male ratio was 77:22. A total of 52 (52.5%) CM patients had underlying conditions, and 47 patients (47.5%) had no underlying conditions. The most common clinical manifestation was headache in 60 patients (60.6%), followed by fever in 51 patients (51.5%), and coma in 8 patients (8.1%). Lumbar punctures were performed at baseline in all patients, the culture of CSF for Cryptococcus was positive for 88 (88%) patients, and India ink preparations were positive for 49 (49%) patients. CSF or blood latex agglutination cryptococcal antigen titer (LACT) of more than 1:512 were demonstrated in 64 (64%) and 62 (62%) patients, respectively. MRI scans detected local lesions in 24 (24%) patients; local lesions were characterized by a low signal in T1, a high signal in T2, and FLARE in MRI, with or without enhancement. Compared to patients without underlying conditions, those with underlying conditions had significantly more fevers, more steroid therapy, higher serum creatinine, and lower albumin, IgG, hemoglobin, and platelets (*p* < 0.05 for all). However, there were no statistically significant differences in gender, age, diabetes, MRI abnormality, CSF examination, and antifungal therapy. The death rate was 23.1% in CM patients with underlying conditions and 21.3% in CM patients without underlying conditions (Table 1). The initial therapy involved intravenous amphotericin B and flucytosine combination for 79 (79%) patients and fluconazole or voriconazole alone for 20 (21%) patients. The K-M survival analysis indicated the mortality was similar between CM patients with and without underlying conditions (Figure 1).

Underlying conditions in the 99 HIV-negative patients who developed CM are listed in Table 1. Kidney transplant represented the most frequent underlying condition (11.1%), followed by rheumatic disease (10.1%), hematological system diseases (9.1%), chronic renal failure (8.1%), and cirrhosis (7.1%). Among the nearly 200,000 hospitalizations coded each year in our hospital during the study period, the total number of patients per underlying condition is shown in Table 2. Data on some conditions were not reliable, either because no specific code exists, because of overlap with other diseases, or because the code was created only recently. Hence, the incidence rates of CM could not be estimated for these conditions.

Environmental exposures and the lifestyle of these patients are shown in Table 3. CM patients without underlying conditions had significantly more alcohol abuse than those with underlying conditions (31.9% vs. 9.6%, *p* = 0.011). However, there were no significant differences between the two groups in terms of occupation, pet ownership, smoking, marijuana use, sleep late, sleep insufficiency, and living in a wet environment.

A total of 22 patients died within 30 days of admission, and the 30-day mortality was 22.2%. In the other 77 patients who were alive 30 days after admission at the time of the last follow-up, the median follow-up time was 44.3 months (range: 1–110 months). Among these 77 patients, 8 patients died after discharge during the follow-up period. Three patients died of reinfection at 5 months, 1 year, and 4 years, respectively, two patients died of heart failure at 8 months, and two patients died of a stroke at 1 month and 4 years, respectively. We also analyzed possible factors associated with mortality between surviving CM patients and deceased CM patients (Table 4). We found that deceased patients were significantly older, more likely to be male, and had higher WBC, CSF WBC, and CSF protein, and lower CSF chloride than surviving patients (*p* < 0.05 for all). By logistic regression analysis, male gender (OR = 3.16, *p* = 0.001), higher CSF WBC (OR = 2.88, *p* = 0.005), and protein (OR = 2.82, *p* = 0.002) were significantly associated with mortality (Table 5).

## 4. Discussion

In this retrospective study, data on 99 HIV-negative patients who developed CM were analyzed. Underlying conditions were present in 52.5% of the CM patients, and 47.5% of patients had no underlying conditions. Kidney transplant represented the most frequent underlying condition (11.1%), followed by rheumatic disease and hematological diseases. These findings are similar to those of some previous studies [9,10]. In a study at 15 medical centers in the United States of 306 HIV-negative patients with *C. neoformans*, 157 had central nervous system involvement, and 79% had a significant underlying condition; the most frequent underlying condition was chronic hepatic failure (16%), followed by solid-organ transplant and hematologic malignancy [9]. In a Chinese study, Zhu et al. reviewed 154 non-HIV-infected patients with CM, the majority of whom were otherwise apparently healthy (66.9%) [10].

The estimated incidence of underlying conditions in non-HIV-infected CM patients in our study varied broadly. The most frequent was kidney transplant (30 cases per 100,000 patient-year), followed by SLE, thrombocytopenic purpura, and primary nephrotic syndrome. Data about the incidence of CM in patients with different underlying conditions is limited. CM in previously healthy individuals is relatively rare, with approximately 3000 cases reported annually in the United States, which would put the incidence at approximately one in 100,000 individuals per year [11]. In two studies of SLE patients with central nervous system infections, the estimated incidence of CM was 13.8–44 cases per 100,000 patient-years [12,13]. Wu et al. examined the incidence and clinical features of CM among different groups of organ transplant patients, and the estimated incidence of CM was 32.9 cases per 100,000 patient-year [14]. These data indicated that patients with kidney transplant and SLE had a higher risk of CM than those with other underlying conditions.

Despite underlying conditions, the mortality of these patients for one month was close to that of patients without underlying conditions in our study. Zhu et al. retrospectively reviewed 154 non-HIV-infected patients with CM and also found that previously healthy patients had a similar treatment response and one-year survival as immunocompromised patients [10]. In a study of 34 HIV-negative children with CM in Beijing, those with no identifiable underlying disease had a similar prognosis both at discharge and follow-up to patients with underlying diseases [15]. In another study, George et al. also found that the mortality rate was similar between HIV-negative CM patients with or without solid organ transplant [16]. These findings indicated that CM without underlying conditions had a higher mortality rate, similar to that of patients with underlying conditions. Maybe the reason that previously healthy patients for whom diagnosis was delayed had more severe disease, and experienced more brain herniation, coma, seizures, hydrocephalus, and more surgical shunt procedures. In addition, post-infectious inflammatory response syndrome (PIIRS) has been reported as a deterioration in neurological status in a previously healthy patient with CM after CSF fungal culture conversion to negative following optimal treatment [17,18].

Although the risk factors for CM in immunocompromised hosts are well described and associated with the net state of immunosuppression, much less is known about the effects of occupation and lifestyle on the risk of CM in the general population. We found that CM patients without underlying conditions had significantly more alcohol abuse than those with underlying conditions, which indicates that alcohol abuse is a probable risk factor for CM in previously healthy patients. Alcohol abuse has been associated with invasive fungal infections, especially in patients with alcoholic cirrhosis [19]. Although no case-control studies have been reported to date, all of the patients that have been described had no other risk factor for invasive fungal infections except that of alcohol abuse. Spontaneous peritonitis due to *C. neoformans* has been described in patients with decompensated alcoholic cirrhosis and ascites [20].

We found that patients who died were significantly older, more likely to be male, had higher WBC, CSF WBC, and CSF protein, and lower CSF chloride than patients who survived. This was similar to some, but not all, previous studies. Guo et al. reviewed 34 HIV-negative children with CM, the patients in the survival group had higher ESR and eosinophil than patients in the death group, and there were no significant differences in age, area, contact history, time to diagnosis, hospitalization, and CSF tests [15]. In another retrospective study, a total of 126 CM and 105 tuberculous meningitis patients were included; multivariate analysis showed that older age, altered mentation, lower CD4/CD8 ratios, and higher CSF cryptococcal antigen were independent risk factors for poor prognosis for CM patients [21]. In an early study, 88 patients with CM were identified, and 37 (42%) were HIV-infected; they found patients who died had significantly lower CSF WBC, but this study contained HIV-positive patients, whom themselves had lower CSF WBC than HIV-negative patients [22]. In a recent study, early clinical and microbiological predictors of outcome in hospitalized patients with CM were evaluated [23], and older age and quantitative CSF yeast counts performed by direct microscopic exam were associated with mortality in multiple analyses by logistic regression [23]. We also found that male gender was significantly associated with mortality. This was not consistent with previous studies, although they also found more male patients in the groups that died compared to the groups that survived (9,10,23). These discrepancies may be due to differences in the characteristics of the studied population.

There were limitations in the present study. First, our study was retrospective research. Usually, there are missing data and misclassification in this kind of study. Second, the sample size was a limiting factor that may have affected the precision of the estimates. Finally, because of some missing data in the long-time follow-up, the long-term risk factors associated with mortality could not be fully determined.

In summary, we described the clinical characteristics and risk factors of HIV-negative CM patients. The main findings were that kidney transplant represented the most frequent underlying condition. Patients with kidney transplant and SLE had a higher risk factor of CM than those with other underlying conditions. Mortality during one month among patients with underlying conditions was similar to that among patients without underlying conditions in our study. Alcohol abuse was a probable risk factor of CM for previously healthy patients, and male gender, higher CSF WBC, and protein were significantly associated with mortality.

## Figures and Tables

**Figure 1 jcm-12-00515-f001:**
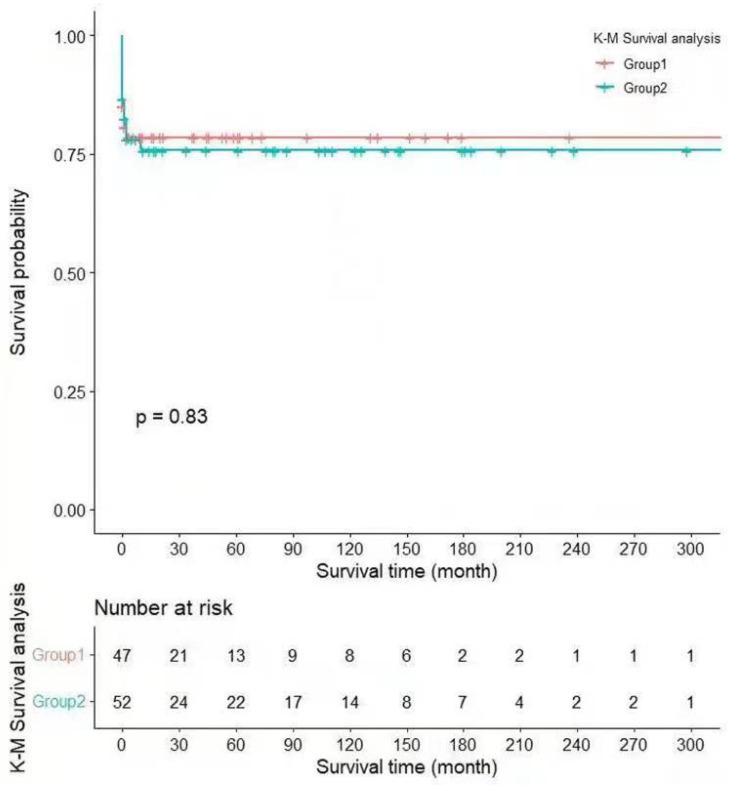
Comparison of the mortality between CM patients with and without underlying conditions. Group 1: Patients without underlying conditions; Group 2: Patients without underlying conditions.

**Table 1 jcm-12-00515-t001:** Clinical characteristics of cryptococcal meningitis (CM) patients with and without underlying conditions (*n* = 99).

Variables	CM with Underlying Conditions (*n* = 52)	CM without Underlying Conditions (*n* = 47)	*p*-Value
Male (*n*, %)	12 (23.1)	10 (21.3)	0.830
Age (years)	57.1 ± 16.2	55.1 ± 16.3	0.535
Headache (*n*, %)	28 (53.8)	32 (68.1)	0.148
Fever (*n*, %)	32 (61.5)	19 (40.4)	0.036 *
Coma (*n*, %)	3 (5.8)	5 (10.6)	0.375
Pulmonary Cryptococcosis (*n*, %)	15 (28.8)	10 (21.3)	0.387
Encephalitis (*n*, %)	3 (5.8)	4 (8.5)	0.595
Cryptococcal septicemia (*n*, %)	6 (11.5)	4 (8.5)	0.618
History of steroid therapy (*n*, %)	35 (67.3)	2 (4.3)	<0.001 *
Diabetes (*n*, %)	12 (23.1)	10(21.3)	0.830
Albumin (g/L)	33.9 ± 6.2	37.5 ± 5.7	0.006 *
IgG (g/L)	9.3 ± 4.0	13.4 ± 10.4	0.017 *
Serum creatinine (μmol/L)	111.4 ± 109.1	62.9 ± 22.0	0.005 *
Hemoglobin (g/dL)	109.6 ± 23.6	126.1 ± 20.5	0.001 *
WBC (10^9^/L)	7.6 ± 4.6	8.0 ± 3.2	0.613
Platelet (10^9^/L)	192.6 ± 87.0	245.2 ± 84.1	0.004 *
ESR (mm/h)	21.5 ± 18.3	20.7 ± 17.8	0.851
CRP (mg/L)	23.3 (1.1–92.9)	19.1 (0.28–107.0)	0.400
Procalcitonin (ng/mL)	0.46 (0–6.5)	0.2 (0–1.5)	0.175
CD4^+^ cell count (cells/mm^3^)	302.0 ± 187.6	309.3 ± 240.7	0.918
CD8^+^ cell count (cells/mm^3^)	490.8 (23–495)	266.8 ± 138.6	0.188
NK cell count (cells/mm^3^)	164.8 ± 157.3	156.6 ± 151.1	0.887
CSF opening pressure (cmH_2_O)	225.2 ± 97.8	240.8 ± 108.7	0.517
CSF WBC (cell/mm^3^)	330.7 (0–9800)	185.2 (0–2160)	0.507
CSF protein (g/L)	2.4 (0.3–33.5)	1.5 (0.1–9.1)	0.329
CSF glucose (mmol/L)	2.7 ± 1.7	2.6 ± 1.4	0.754
CSF chloride (mmol/L)	117.3 ± 6.9	116.1 ± 7.2	0.424
MRI abnormality (*n*, %)	11 (21.2)	13 (27.7)	0.451
Therapy			
Amphotericin B + flucytosine (*n*, %)	38 (73.1)	41 (87.2)	0.080
Fluconazole or Voriconazole (*n*, %)	14 (26.9)	6 (12.8)	0.080
Died (%)	12 (23.1)	10 (21.3)	0.830

WBC, white blood cell; ESR, erythrocyte sedimentation rate; CRP, C-reactive protein; CSF: cerebrospinal fluid; MRI: magnetic resonance imaging. Data are shown as the mean ± SD (standard deviation) or the percentages. * *p* < 0.05.

**Table 2 jcm-12-00515-t002:** Main underlying conditions in 99 consecutive cases of cryptococcal meningitis (CM) in non-HIV-infected patients, and the number of patients hospitalized with this condition (2010–2021).

Underlying Condition	Number of CM (% of Total Cases in Non-HIV Patients)	Total Number of Patients with the Underlying Condition
Kidney transplant	11 (11.1%)	3048
Rheumatic disease	10 (10.1%)	
Systemic lupus erythematosus	8	6934
Rheumatoid arthritis	1	11828
Giant cell arteritis	1	22
Hematological system diseases	9 (9.1%)	
Leukemia	3	6170
Lymphoma	1	8412
Multiple myeloma	1	2086
Myelodysplastic syndrome	1	1788
Thrombocytopenic purpura	2	2421
Autoimmune hemolysis	1	2086
Primary nephrotic syndrome	5 (5%)	8277
Cirrhosis	7 (7.1%)	
Chronic renal failure	8 (8.1%)	
Solid tumors	2 (2.0%)	
Breast cancer	1	
Thymoma	1	
None	47 (47.0%)	

**Table 3 jcm-12-00515-t003:** Environmental exposures and lifestyle of CM patients with and without underlying conditions (*n* = 99).

Variables	CM with Underlying Conditions (*n* = 52)	CM without Underlying Conditions (*n* = 47)	*p*-Value
Farmer (*n*, %)	3 (5.8)	4 (8.5)	0.705
Construction workers (*n*, %)	2 (3.8)	4 (8.5)	0.419
Landscapers (*n*, %)	1 (1.9)	2 (4.3)	0.603
Pet ownership (*n*, %)	3 (5.8)	4 (8.5)	0.705
Smoking (*n*, %)	10 (19.2)	12 (25.5)	0.478
Alcohol abuse (*n*, %)	5 (9.6)	15 (31.9)	0.011 *
Marijuana use (*n*, %)	1 (1.9)	1 (2.1)	1.000
Sleep late (*n*, %)	15 (28.8)	19 (40.4)	0.290
Sleep insufficiency (*n*, %)	21 (40.4)	26 (55.3)	0.161
Living in wet environment (*n*, %)	5(9.6)	8 (17.0)	0.374
Tourism in the past 3 months (*n*, %)	1 (1.9)	2 (4.3)	0.603

CM, cryptococcal meningitis; * *p* < 0.05.

**Table 4 jcm-12-00515-t004:** Possible risk factors associated with mortality between surviving CM patients and deceased CM patients (*n* = 99).

Variables	Surviving Patients (*n* = 77)	Deceased Patients (*n* = 22)	*p*-Value
Male (*n*, %)	10 (13.0)	12 (54.5)	<0.001 *
Age	54.5 ± 15.8	62.1 ± 16.5	0.043 *
Coma (*n*, %)	5 (6.5)	3 (13.6)	0.278
Pulmonary cryptococcosis (*n*, %)	21 (27.3)	4 (18.2)	0.387
Encephalitis (*n*, %)	4 (5.2)	3 (13.6)	0.173
Cryptococcal septicemia (*n*, %)	7 (9.1)	3 (13.6)	0.533
History of steroid therapy (*n*, %)	30 (39.0)	7 (31.8)	0.541
Underlying conditions (*n*, %)	43 (55.8)	9 (40.9)	0.216
Kidney transplant (*n*, %)	11 (14.3)	0 (0)	0.060
Rheumatic disease (*n*, %)	7 (9.1)	3 (13.6)	0.533
Hematological diseases (*n*, %)	8 (10.4)	1 (4.5)	0.400
Primary nephrotic syndrome (*n*, %)	4 (5.1)	1 (4.5)	0.921
Chronic hepatic failure (*n*, %)	4 (5.1)	2 (9.1)	0.480
Chronic renal failure (*n*, %)	5 (6.5)	2 (9.1)	0.675
Albumin (g/L)	35.6 ± 6.6	35.1 ± 4.9	0.739
IgG (g/L)	10.9 ± 7.8	9.7 ± 2.4	0.565
Serum creatinine (μmol/L)	98.3 ± 93.8	57.5 ± 22.7	0.052
Hemoglobin (g/dL)	116.6 ± 23.5	119.0 ± 24.1	0.678
WBC (10^9^/L)	7.3 ± 3.5	9.4 ± 5.3	0.042 *
Platelet (10^9^/L)	216.5 ± 81.9	217.2 ± 113.4	0.973
ESR (mm/h)	21.2 ± 18.8	21.0 ± 13.8	0.973
CRP (mg/L)	19.4 (0.28–107)	27.6 ± 22.9	0.182
Procalcitonin (ng/mL)	0.33 (0–6.5)	0.42 (0–3.1)	0.718
CD4^+^ cell count (cells/mm^3^)	407.6 ± 327.2	180 ± 177.7	0.305
CD8^+^ cell count (cells/mm^3^)	431.7 ± 392.1	247.5 ± 110.6	0.467
NK cell count (cells/mm^3^)	171.2 ± 160.2	94 ± 90.4	0.360
CSF opening pressure (cmH_2_O)	227.7 ± 96.4	260.5 ± 133.6	0.317
CSF WBC (cell/mm^3^)	138.2 (0–2160)	774.6 (0–9800)	0.019 *
CSF protein (g/L)	1.4 ± 1.2	4.4 (0.1–33.5)	0.008 *
CSF glucose (mmol/L)	2.7 ± 1.2	2.5 ± 1.6	0.631
CSF chloride (mmol/L)	117.6 ± 7.2	112.8 ± 4.6	0.011 *
MRI abnormality (*n*, %)	20 (26.0)	4 (18.2)	0.452
Therapy			
Amphotericin B + flucytosine (*n*, %)	61 (79.2)	18 (81.8)	0.789
Fluconazole or Voriconazole (*n*, %)	16 (20.8)	4 (18.2)	0.789

CM, cryptococcal meningitis; WBC, white blood cell; ESR, erythrocyte sedimentation rate; CRP, C-reactive protein; CSF: cerebrospinal fluid; MRI: magnetic resonance imaging. Data are shown as the mean ± SD (standard deviation) or the percentages. * *p* < 0.05.

**Table 5 jcm-12-00515-t005:** Logistic regression analysis with outcomes as the dependent variable (R^2^ = 0.68). OR, odds ratios; CI, confidence interval; cerebrospinal fluid; WBC, white blood cell; CSF.

Variable	OR	95% CI for OR	*p*-Value
Male	3.16	2.11–12.63	0.001
CSF WBC	2.88	1.60–14.90	0.005
CSF protein	2.82	1.82–12.79	0.002

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
