# Peer review of "Cryptococcal Meningitis in HIV-Negative Patients: A 12-Year Single-Center Experience in China"

_jcm, 2023, doi:10.3390/jcm12020515_

Round 1

Reviewer 1 Report

Huang and colleagues performed a retrospettive study about CM meningitis in HIV negative patients. The topic is of course interesting and data are needed to better describe this disease.

Abstract line 19-23: this part is a bit ripetitive, you can simply report the frequency of underlying conditions omitting the patient-year data.

Introduction. You can expand this section, for example you can add the most common comorbidities in HIV-negative patients and that a significant proportion of patients has no underlying conditions. Line 51-53: in the last paragraph of the Introduction please report the study design (ie, "We performed a retrospettive observational study to descrive incidence, risk factors...").

Methods. You can expand this section: describe methods of CSF culture and India Ink. Were CSF criptococcal antigen or PCR performed? What about serum antigen? Did you repeat lumbar tap At scheduled time? There was a protocol for brain imaging?

Results. I woul re-organize this section. In the first part you should describe the general characteristics of Your entire cohort of patients (Table 2 should be Table 1, and global data should be reported).Then, you discuss The differences between patients with and without comorbidities (Table1 should be Table 2) and you can presente the KM curve. I would delete Figure 2, it is not really useful.

You can better describe microbiological characteristics (frequency of positive CSF culture and antigen) and also MRI (the most common alterations). In some studies (ie ref 9-10) The Authors used histopathological data to confirm CM diagnosis. Do you have such data?

Please describe treatment of these patients. 

Statistics. Considering that this is a retrospective study, incidence cannot be estimated. Data expressed as per 100000 patients days should not be reported. 

Author Response

Reviewer 1:

Huang and colleagues performed a retrospettive study about CM meningitis in HIV negative patients. The topic is of course interesting and data are needed to better describe this disease.

Abstract line 19-23: this part is a bit ripetitive, you can simply report the frequency of underlying conditions omitting the patient-year data.

We have deleted the patient-year data.

Introduction. You can expand this section, for example you can add the most common comorbidities in HIV-negative patients and that a significant proportion of patients has no underlying conditions. Line 51-53: in the last paragraph of the Introduction please report the study design (ie, "We performed a retrospettive observational study to descrive incidence, risk factors...").

We have expanded this section and added the study design.

Methods. You can expand this section: describe methods of CSF culture and India Ink. Were CSF criptococcal antigen or PCR performed? What about serum antigen? Did you repeat lumbar tap At scheduled time? There was a protocol for brain imaging?

We have expanded this section as your comments.

Results. I woul re-organize this section. In the first part you should describe the general characteristics of Your entire cohort of patients (Table 2 should be Table 1, and global data should be reported).Then, you discuss The differences between patients with and without comorbidities (Table1 should be Table 2) and you can presente the KM curve. I would delete Figure 2, it is not really useful.

We have re-organized this section as your comments.

You can better describe microbiological characteristics (frequency of positive CSF culture and antigen) and also MRI (the most common alterations). In some studies (ie ref 9-10) The Authors used histopathological data to confirm CM diagnosis. Do you have such data?

We have added the microbiological characteristics and MRI in the method part, however, there was no  histopathological data in our patients.

Please describe treatment of these patients.

We have added in the methods part.

Statistics. Considering that this is a retrospective study, incidence cannot be estimated. Data expressed as per 100000 patients days should not be reported.

We have deleted it.

Reviewer 2 Report

Thank you for inviting me to review this manuscript. I have some comments that could be of use:

·      Methods: Please add a paragraph regarding microbiological techniques for the isolation and identification of Cryptococcus spp.

·      Please define what you mean by ‘renal failure’. Is it per some specific guideline? Is it per some specific GFR?

·      English needs minor revision

·      Line 112: please rephrase. The phrase ‘had more fever’ just sounds wrong. Probably, those characteristics were more frequent in that group

·      Line 115: do you mean rate of treatment with antifungals? Or type of treatment?

·      Line 120: See line 112. More alcohol use? Do you mean that it was more frequent? Or of a higher amount??

·      You could put the figure and tables in the text, not after. I think that MDPI journals encourage that

·      Table 1: Why are some numbers followed by a parenthesis and others are not?

·      Tables: You could add a space between the number and the parenthesis

·      I didn’t understand whether the logistic regression was multivariate or not. If yes, state it. If not, it would be of help to perform a multivariate logistic regression analysis to identify factors independently associated with the outcome

·      I am confused with the results of the binary logistic regression analysis. How come the p-values are <0.05 but 95% of CIs contain 1 in two out of the three parameters?

·      A paragraph mentioning the limitations of the study should be added before the conclusion subsection of the discussion section. For example, this is a single-center study with a relatively small number of patients

Author Response

Reviewer 2:

Thank you for inviting me to review this manuscript. I have some comments that could be of use:

Methods: Please add a paragraph regarding microbiological techniques for the isolation and identification of Cryptococcus spp.

We have added it.

Please define what you mean by ‘renal failure’. Is it per some specific guideline? Is it per some specific GFR?

We have added it.

English needs minor revision.

We have improved by a native English speaker.

Line 112: please rephrase. The phrase ‘had more fever’ just sounds wrong. Probably, those characteristics were more frequent in that group.

We have changed it.

Line 115: do you mean rate of treatment with antifungals? Or type of treatment?

We mean the type of treatment, and we added a sentence in this part.

Line 120: See line 112. More alcohol use? Do you mean that it was more frequent? Or of a higher amount??

We mean more frequent,we defined it in the method part.

You could put the figure and tables in the text, not after. I think that MDPI journals encourage that.

We will re-typeset it in the last version.

Table 1: Why are some numbers followed by a parenthesis and others are not?

The percent in parentheses mean total of this diseases.

Tables: You could add a space between the number and the parenthesis

We have changed it.

I didn’t understand whether the logistic regression was multivariate or not. If yes, state it. If not, it would be of help to perform a multivariate logistic regression analysis to identify factors independently associated with the outcome.

Dear reviewer: logistic regression was multivariate, we have explained in the text.

I am confused with the results of the binary logistic regression analysis. How come the p-values are <0.05 but 95% of CIs contain 1 in two out of the three parameters?

Dear reviewer: there had some mistakes in this part because of negligence of authors, we checked the statistics results carefully and recounted it, the final result was shown in table 5.

A paragraph mentioning the limitations of the study should be added before the conclusion subsection of the discussion section. For example, this is a single-center study with a relatively small number of patients.

We have added a paragraph before the discussion section.

Round 2

Reviewer 1 Report

The Authors have update the manuscript according to Suggestions. 

Reviewer 2 Report

The manuscript has been improved during the revision process.